# Oral creatine in hemodialysis patients increases physical functional capacity and muscle mass, an open label study

Waldo Bernales-Delmon[1,2]*, Simón Schulz[3], Iván Guglielmi[4], Cynthia Saravia[4], Yasna Venegas[4], Jaime Joost[5], José Aguilar[5], Andrés Wulf[2], Paulina Bittner[2], María Claudia Martínez[2], Sandy Gómez[2], Catalina Chávez[2], Juan John[2], Felipe Matus[2], Carla Basualto-Alarcón[4,6,7¤]*

**1** Nephrology Service, Hospital Regional Coyhaique, Aysén, Chile, **2** Dialysis Unit, Hospital de Puerto Aysén, Aysén, Chile, **3** Section for Oral Health, Heidelberg Institute of Global Health, Heidelberg, Germany, **4** PUENTES Center for Health Research, Health Sciences Department, University of Aysén, Coyhaique, Aysén, Chile, **5** Rehabilitation Unit, Hospital de Puerto Aysén, Aysén, Chile, **6** Laboratory of Cellular Physiology and Metabolism, Health Sciences Department, University of Aysén, Coyhaique, Aysén, Chile, **7** Anatomy and Legal Medicine Department, Faculty of Medicine, University of Chile, Santiago, Metropolitan Region, Chile

¤ Current address: Department of Health Sciences, University of Aysén, Coyhaique, Aysén Region, Chile
* carla.basualto@uaysen.cl (CBA); wjbernalesd@hotmail.com (WBD)

## Abstract

### Background and hypothesis

Individuals undergoing chronic hemodialysis represent a population with high morbidity and mortality, primarily due to poor nutritional status, chronic inflammation, and cardiovascular disease. However, additional factors, such as low physical activity and impaired functionality, have also been identified as directly associated with increased mortality.

### Main objective

This study was conceived as a pilot study to investigate whether creatine supplementation (5g/day) for eight weeks could provide benefits in terms of physical functionality, handgrip and body composition in a group of adult patients on chronic hemodialysis. On dialysis days, creatine was administered immediately post-dialysis, while on non-dialysis days, patients took the supplement at home. Measurements were taken using bioimpedance analysis, handgrip strength (via dynamometry), and the Short Physical Performance Battery (SPPB), both before starting creatine supplementation and at week 8 of treatment.

### Results

After performing robust statistical analysis, following creatine supplementation, an increase in SPPB scores was observed, with a mean improvement of 0.78 points

**Data availability statement:** All relevant data are within the manuscript and its Supporting Information files (Data table supplied as Excel file).

**Funding:** CBA received MINEDUC (Ministerio de Educación) URY 20993 and URY 21991 funding. https://www.mineduc.cl/ The funders had no role in study design, data collection and analysis, decision to publish, or preparation of the manuscript.

**Competing interests:** The authors have declared that no competing interests exist.

[95% CI: 0.17–1.44] and an effect size of 0.53. Skeletal muscle mass increased by an average of 1.31 kg [95% CI: 0.55 to 2.23], with an effect size of 0.66. Fat-free mass showed a mean increase of 2.11 kg [95% CI: 0.75 to 3.58] with an effect size of 0.64, while phase angle rose by 0.52 degrees [95% CI: 0.27 to 0.76], corresponding to an effect size of 0.90. Regarding volumetric estimates, total body water increased by 1.17 L [95% CI: 0.26 to 2.13] with an effect size of 0.54, and intracellular water increased by 0.97 L [95% CI: 0.48 to 1.51] with an effect size of 0.81. No significant differences were observed in extracellular water with change of 0.20 L [95% CI: −0.30 to 0.70] or handgrip strength with an increment of 0.67 kgF, [95% CI: −0.67 to 2.11].

## Conclusion

Oral creatine supplementation in HD patients for eight weeks improved muscular and functional outcomes and may be proposed as a strategy to mitigate the elevated morbidity observed in this group of patients.

---

## Introduction

Chronic kidney disease (CKD) is a significant cause of mortality worldwide [1]. In its most advanced stage (stage 5), CKD patients require extracorporeal support therapies such as hemodialysis (HD) [2]. This group of individuals on dialysis represents a particularly vulnerable population, exhibiting higher morbidity and mortality rates when compared to the general population [3]. Among the most well-known determinants of this unfavorable outcome are: the presence of multiple comorbidities, chronic inflammation, anemia, protein-energy-wasting, and a significant burden of cardiovascular disease [4,5]. However, other determinants are often overlooked, such as low functionality, (with a prevalence of approximately 50% in this population) [6], and decreased physical activity [7], both of which are strongly related to dependence and mortality [8,9]. In the general population, multiple studies have linked creatine supplementation to increased strength and performance in association to exercise [10,11]. In patients with kidney disease, especially in its most advanced stage, the use of creatine has been less studied. Specifically in the hemodialysis population, studies have revealed increased creatine losses during dialysis sessions, primarily from intracellular reserves. At the same time, low creatine levels have been related with increased fatigability, reduced muscle mass, hypoalbuminemia, and low protein intake [12,13]. Following this hypothesis, some pilot studies on intradialytic creatine supplementation have been designed, but no results have been reported to date [14].

Therefore, the objective of this pilot research was to determine whether oral creatine supplementation improves physical functional performance and muscle mass in a convenience sample of adults dialysis patients, measured before and after creatine supplementation for eight weeks.

## Materials and methods

### Trial design

This is an open label single-arm repeated-measures feasibility clinical trial.

### Patient selection

A convenience sampling of patients at "Hospital de Puerto Aysén Hemodialysis Unit" was initially recruited for this pilot study, primarily based on their good adherence to pharmacological and dialysis therapy. Patient recruitment began on October 9th and concluded on October 30st, 2023.

Efforts were made to ensure that the average age of the selected group matched the overall average age of all patients attending therapy in the unit.

The remaining inclusion criteria were the following: 18 years or older, having started dialysis at least three months before the start of supplementation, not being on any formal exercise program and not using any nutritional supplement. Individuals with pacemakers or any type of metal prosthesis were excluded. All the recruited patients received creatine supplementation. They were measured before (pre-) and after (post-) receiving creatine supplementation.

### Interventions procedures

**Creatine supplementation.** For creatine supplementation, we administered 5 g/day of Creapure™ monohydrate creatine powder, dissolved in 100 milliliters of water. On dialysis days, the supplementation was provided immediately after the session. On non-dialysis days, the 5 g dose of creatine powder was dispensed in pre-prepared containers by the dialysis unit nutritionist, and each patient received instructions on how to prepare the supplement at home using 100 milliliters of water. All patients were aware that they were receiving creatine, and all of them received continuous creatine supplementation for eight weeks.

**Outcomes.**

**Primary outcome:** To compare the physical performance status of selected patients, as assessed by the Short Physical Performance Battery (SPPB), before and after eight weeks of creatine supplementation.

**Secondary outcomes:** To compare changes in muscle strength, measured using a hand dynamometer, and changes in body composition, estimated by bioimpedance analysis (fat free mass, skeletal muscle mass, total body water, extracellular water, and intracellular water). Additionally, to evaluate changes in phase angle.

All comparisons were made before and after eight weeks of creatine supplementation.

### Laboratory tests

The laboratory tests included in the analysis were part of the routine monthly examinations performed on all patients enrolled in our units hemodialysis program. These tests are conducted at the beginning of each month, specifically pre- and post-dialysis during the mid-week session, in accordance with international guidelines [15].

For this study, we analyzed levels of calcium, phosphorus, blood urea nitrogen, hematocrit, albumin, serum creatinine, and vitamin D. The Kt/V was calculated using the second-generation Daugirdas formula [16], and the normalized protein catabolic rate (nPCR) was calculated using the following formula [17]:

$$nPCR = 0.0136 \; x \; \left( \frac{Kt}{V} x \; \left( \frac{pre-dialysis\;BUN \; + \; post-dialysis\;BUN}{2} \right) \right) \; + 0.251$$

Pre-supplementation laboratory tests were conducted one week before the initiation of creatine supplementation. The second round of tests were conducted during supplementation, specifically finishing week eight.

Biochemical analyses were performed using the ARCHITECT ci4100 analyzer (Abbott), and hematological analyses were conducted using the CELL-DYN Ruby analyzer (Abbott).

## Bioimpedance analysis

Measurements were taken using the SECA 525™ bioimpedance analyzer, following the manufacturers instructions. Electrodes were placed on both hands and both feet. Measurements were conducted in the morning after the second dialysis session of the week. Thus, for patients undergoing dialysis on Mondays, Wednesdays, and Fridays, measurements were taken on Thursday, while for those dialyzed on Tuesdays, Thursdays, and Saturdays, measurements were conducted on Friday. The same schedule was followed for both pre- and post-supplementation measurements. All assessments were conducted in a fasting state, with participants advised to refrain from strenuous exercise for at least two hours beforehand. Notably, all measurements were obtained in the early morning hours, prior to 10:00 a.m.

The analyzed variables included total body water, extracellular water, intracellular water (calculated by subtracting extracellular water from total body water), skeletal muscle mass, fat-free mass, and phase angle. Phase angle is a parameter related to quantity and quality of soft tissues, with higher values reflecting greater cellularity, improved cellular integrity, and enhanced cellular functionality [18–20]. The only variable calculated by the investigators was intracellular water; all other variables were directly provided by the bioimpedance device.

## Functional and strength tests

Functional performance was assessed using the Short Physical Performance Battery (SPPB), which comprises three sections scored separately on a scale from 0 to 4 points. The first section evaluates balance, the second assesses gait speed, and the third measures the ability to stand up and sit down from a chair five times without using the arms. Each section has a maximum score of 4 points, with the overall score ranging from 0 to 12 [21].

For handgrip strength measurement, three grip strength attempts were conducted on each hand using a hydraulic hand dynamometer (Baseline Lite™), with a one-minute rest between attempts. The highest value of the three measurements, regardless of whether the stronger hand was on the same side of the fistula, was used for analysis.

Both the SPPB and strength tests were performed and supervised by two physical therapists from the hospitals rehabilitation and exercise team, where the dialysis unit is located. These measurements were conducted on the same day, immediately following the bioimpedance analysis.

Dry weight was clinically estimated based primarily on physical examination, blood pressure values, and patient-reported symptoms.

## Sample size calculation

Due to the lack of studies investigating changes in SPPB with creatine supplementation in dialysis patients, we relied on data from studies examining the effects of exercise on this population and the corresponding changes in SPPB scores. Based on these studies, we considered a conservative effect size of 0.6 [22]. Therefore, the required sample size for a two-tailed test comparing means in two dependent samples, with an alpha level of 0.05, a power of 0.8, and an effect size of 0.6, was 25 patients. Due to the restricted number of dialysis patients in our unit (n: 56), we were only able to approach the calculated sample size (more details in the results section). The size calculation was made using G*POWER 3.1 software.

## Data analysis procedures

In order to mitigate the limitations of not reaching our calculated sample size we decided to analyze our data by using a robust statistics approach. This methodology is less sensitive to sample size, outliers or deviations from idealized statistical models, thus allowing for more reliable results.

To determine whether there were differences in the variables after creatine supplementation, we estimated the mean differences between before and after the intervention using a bootstrap analysis with 10,000 resample. This method was

used to detect statistically significant changes in variables related to both the primary and secondary outcomes, as well as in exploratory variables obtained from laboratory tests performed on the participants. Results are presented as mean differences, standard deviations of those differences, and 95% confidence intervals. For the primary and secondary outcome variables, effect sizes were also calculated using Hedges' *g*, which is considered more appropriate than Cohen's *d* for small sample sizes (n < 20) [23]. Categorical variables are shown as raw number and percentage. Continuous variables are shown as mean and standard deviation (SD).

All statistical analyses and figures were conducted using R 4.5.0 and Jamovi desktop v2.6.26

### Ethics committee

The research protocol was approved by the local ethics committee under Ordinance Number 25, dated October 2nd, 2023. The ethics committee is part of the Servicio de Salud Aysén del General Carlos Ibáñez del Campo.

All participants provided written informed consent.

## Results

### Baseline characteristics

Out of a total of fifty-six patients in our dialysis unit, we were initially able to recruit only nineteen. This limitation was due to a high percentage of patients experiencing difficulties with independent mobility, which prevented them from completing the SPPB test. Additionally, some patients lacked a strong support network at home, making it impossible to ensure consistent creatine use, while others demonstrated poor adherence to their pharmacological therapy. From this initial selection, we had to withdraw one patient because he failed to attend measurements on the scheduled dates and didn't follow creatine supplementation instructions. Of the remaining eighteen patients nine (50%) were female, the participants mean age was 59.3 years (11.6), and an average dialysis vintage of 61 months (78.9) (Table 1). The majority of patients, twelve (66%), were undergoing high-flux hemodialysis, while six (33%) were on online hemodiafiltration, with each patient maintaining their respective modality throughout the intervention period.

Regarding underlying pathologies, sixteen (88%) patients were diagnosed with arterial hypertension, ten (52.6%) were diabetic, and three (15.7%) presented with vascular disease, defined as a history of coronary heart disease, cerebrovascular accident, or peripheral arterial occlusive disease. In terms of vascular access, fourteen (73.6%) patients had native arteriovenous fistulas, one (11.1%) had a prosthetic arteriovenous fistula, and three (15.7%) had tunneled venous catheters. Notably, ten (57.8%) patients exhibited diuresis of less than 0.2 liters per day. The dietary intake or physical activity were not recorded during the study. Throughout the course of the trial, all patients received the same routine nutritional counseling as the other patients in the dialysis unit. This counseling was provided by the units permanent staff dietitian who also looked after creatine negative side effects; patients did not report to experience neither bloating or nausea. Also there was no interdialytic weight gain due to changes in water volume.

The laboratory examination results described in Table 1 did not demonstrate significant variations pre- and post-creatine supplementation, with the exception of serum creatinine levels, which increased by 2.01 mg/dL ± 1.19 [95% CI: 1.45 to 2.56] following supplementation, and vitamin D levels, which were higher by 6.63 ng/ml ± 12.9 [95% CI: 1.11 to 12.98] in the post-creatine supplementation measurement. It is important to consider that patients were supplemented with vitamin D as part of the standard treatment for dialysis patients, and that pre-supplementation measurements were conducted in winter, while post-intervention measurements were performed during the summer period.

Indicators of dialysis adequacy and therapy remained stable following creatine supplementation. The single-pool Kt/V showed a mean change of 0.04 ± 0.30 [95% CI: −0.06 to 0.18], while dry weight exhibited a non-significant mean change of 0.09 kg ± 3.9 [95% CI: −1.89 to 1.61]. Nutritional indicators, such as nPCR, which showed a mean change of −0.04 ± 0.3 [95% CI: −0.10 to 0.16], and serum albumin, with a mean change of 0.14 mg/dL ± 0.3 [95% CI: −0.27 to 0.04], also remained unchanged between the pre- and post-intervention assessments.

**Table 1. Participant's clinical characteristics.**

| Characteristic | Pre (n = 18) | Post (n = 18) | [95% CI][1] |
|---|---|---|---|
| **Women, n (%)** | 9 (50%) | --- | |
| **Age, years** | 59.3 (11.6) | --- | |
| **Dialysis Vintage, months** | 61 (78.9) | --- | |
| **Renal Replacement Therapy Modality, n (%)** | | | |
| High Flux Hemodialysis | 12 (66.6%) | --- | |
| Hemodiafiltration | 6 (33.3%) | --- | |
| **Comorbidities, n (%)** | | | |
| Hypertension | 16 (88.8%) | --- | |
| Diabetes | 10 (52.6%) | --- | |
| Vascular disease # | 3 (15.7%) | --- | |
| **Vascular Access, n (%)** | | | |
| AVF | 14 (73.6%) | --- | |
| AVG | 1 (11.1%) | --- | |
| Tunneled CVC | 3 (15.7%) | --- | |
| **Diuresis < 0.2 L/day, n (%)** | 10 (57.8%) | --- | |
| **Hematocrit %** | 35.7 (4.0) | 33.3 (4.4) | [-5.12 to 0.98] |
| **Calcium mg/dL** | 7.9 (0.6) | 8.1 (0.6) | [-0.22 to 0.42] |
| **Phosphorus mg/dL** | 4.9 (1.8) | 4.9 (1.0) | [-0.62 to 0.71] |
| **Blood Urea Nitrogen mg/dL** | 65.9 (18.9) | 70.3 (16.0) | [-4.44 to 12.83] |
| **Creatinine mg/dL** | 8.5 (2.3) | 10.5 (2.6) | [1.45 to 2.56]* |
| **Vitamin D ng/mL** | 22.4 (10.5) | 30.8 (10.8) | [1.11 to 12.98]* |
| **Serum Albumin g/dL** | 3.8 (0.29) | 3.7 (0.25) | [-0.27 to 0.04] |
| **nPCR g/kg/day** | 1.2 (0.32) | 1.2 (0.25) | [-0.10 to 0.16] |
| **Dialysis Dose, Kt/V single pool** | 1.65 (0.28) | 1.69 (0.34) | [-0.06 to 0.18] |
| **Stature meters** | 1.56 (0.09) | --- | |
| **Dry Weight kg** | 76.2 (11.7) | 76.1 (12.7) | [-1.89 to 1.61] |

[1]95% confidence interval based on 10 000 bootstrap replicates of the pre–post differences.

*Statistically significant at 5% level.

#Coronary artery disease, peripheral vascular disease or stroke; AVF: arteriovenous fistula; AVG: arteriovenous graft; CVC: central venous catheter.
Categorical variables are shown as raw number and percentage. Continuous variables are shown as mean and standard deviation (SD).

## Functionality and muscle mass improve following creatine supplementation

The application of the SPPB test objectively demonstrated a significant increase in scores, rising from a mean of 10.33 ± 2.32 to 11.11 ± 1.49, with a mean difference of 0.78 ± 1.40 [95% CI: 0.17 to 1.44], following 8 weeks of creatine supplementation (Table 2 and Fig 1A). The calculated effect size was 0.53 (Hedges' g), indicating a medium effect.

Muscle strength showed a slight, non-significant increase, from a mean of 28.28 ± 9.42 kgF to 28.94 ± 10.11 kgF, with a mean difference of 0.67 ± 3.11 kgF [95% CI: –0.67 to 2.11]. The effect size was 0.21, indicating a small effect. (Table 2 and Fig 1B)

According to bioimpedance analysis (Table 2 and Fig 2), post-supplementation measurements revealed significant changes in the following variables: Skeletal muscle mass increased by 1.31 ± 1.89 kg [95% CI: 0.55 to 2.23], rising from 21.36 ± 5.28 kg to 22.67 ± 5.45 kg, with an effect size of 0.66, indicating a moderate effect (Fig 2A). Fat-free mass increased by 2.11 ± 3.16 kg [95% CI: 0.75 to 3.58], with an effect size of 0.64, increasing from 44.89 ± 10.11 kg

**Table 2. Functionality test, handgrip strength and bioimpedanciometry Bootstrap analysis pre- and post creatine supplementation.**

| Parameter | Pre-Creatine | Post-Creatine | Mean Difference [95% CI][1] | Hedges'g [95% CI][2] |
|---|---|---|---|---|
| SPPB (points) | 10.33 (2.33) | 11.11 (1.49) | 0.78 [0.17 to 1.44]* | 0.53 [0.13 to 1.01]* |
| Strength (kgF) | 28.28 (9.42) | 28.94 (10.11) | 0.67 [−0.67 to 2.11] | 0.21 [- 0.31 to 0.62] |
| **Bioimpedanciometry** | | | | |
| Skeletal Muscle Mass (kg) | 21.36 (5.29) | 22.67 (5.46) | 1.31 [0.55 to 2.23]* | 0.66 [0.39 to 1.20]* |
| Fat-Free Mass (kg) | 44.89 (10.85) | 47.01 (11.17) | 2.11 [0.75 to 3.58]* | 0.64 [0.27 to 1.18]* |
| Total Body Water (liters) | 35.37 (7.07) | 36.53 (7.12) | 1.17 [0.26 to 2.13]* | 0.54 [0.14 to 1.12]* |
| Extracellular Water (liters) | 16.48 (2.96) | 16.68 (2.98) | 0.20 [−0.30 to 0.70] | 0.17 [- 0.28 to 0.71] |
| Intracellular Water (liters) | 18.88 (4.43) | 19.85 (4.44) | 0.97 [0.48 to 1.51]* | 0.81 [0.46 to 1.41]* |
| Phase Angle (degrees) | 5.36 (1.01) | 5.88 (1.04) | 0.52 [0.27 to 0.76]* | 0.90 [0.46 to 1.73]* |

[1]Point estimate and 95% confidence interval based on 10 000 bootstrap replicates of the pre–post differences.

[2]Effect size estimates with 95% confidence intervals based on 10,000 bootstrap resamples.

*Statistically significant at 5% level.

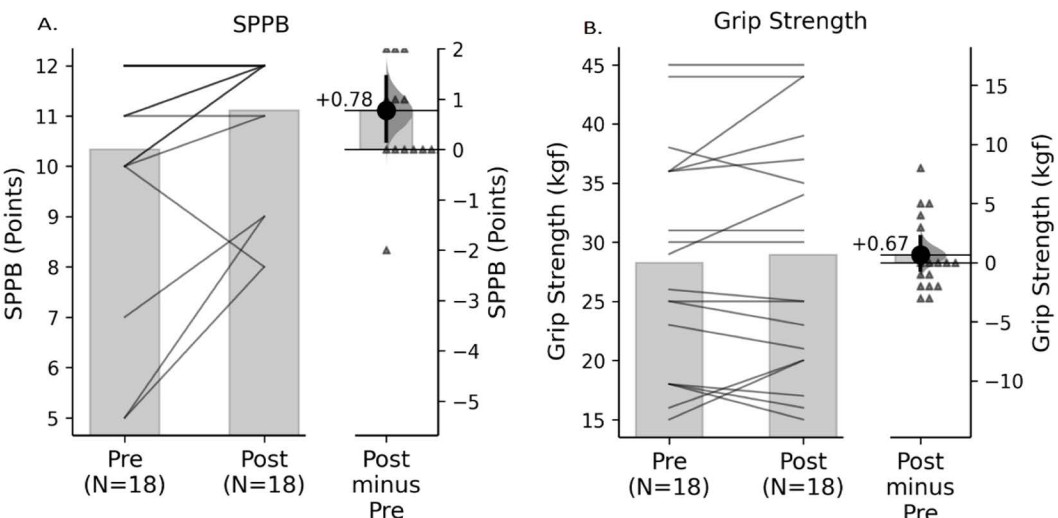

**Fig 1. Gardner-Altman plots showing changes from pre- to post-intervention. Physical performance (SPPB) but not handgrip strength, increases after creatine supplementation.** Strength measured as handgrip (B) did not change after creatine supplementation. Functional capacity, measured through SPPB test (A), showed significant increases after 8 weeks of oral creatine supplementation. The left panel displays paired individual values, with lines connecting pre- and post-intervention measurements. The right panel illustrates the mean difference as a dot, with the 95% confidence interval indicated by the vertical bar. Positive values denote improvement after the intervention.

to 47.01 ± 10.05 kg (Fig 2B). Phase angle showed a significant increase of 0.52 ± 0.55 degrees [95% CI: 0.27 to 0.76], rising from a baseline mean of 5.36 ± 1.01 degrees to 5.88 ± 1.04 degrees, with an effect size of 0.90, indicating a large effect (Fig 2F).

Regarding volumetric variables, significant changes were observed in total body water, which increased by 1.17 L ± 2.07 [95% CI: 0.26 to 2.13], changing from 35.37 L ± 7.07 L to 36.53 L ± 7.12 L. The calculated effect size was 0.54, indicating a moderate effect (Fig 2C). Importantly, the increase in total body water was mainly driven by a rise of 0.97 L ± 1.14 [95% CI: 0.48 to 1.51] in intracellular water changing from 18.88 L ± 4.43 L to 19.85 L ± 4.44 L, following creatine supplementation,

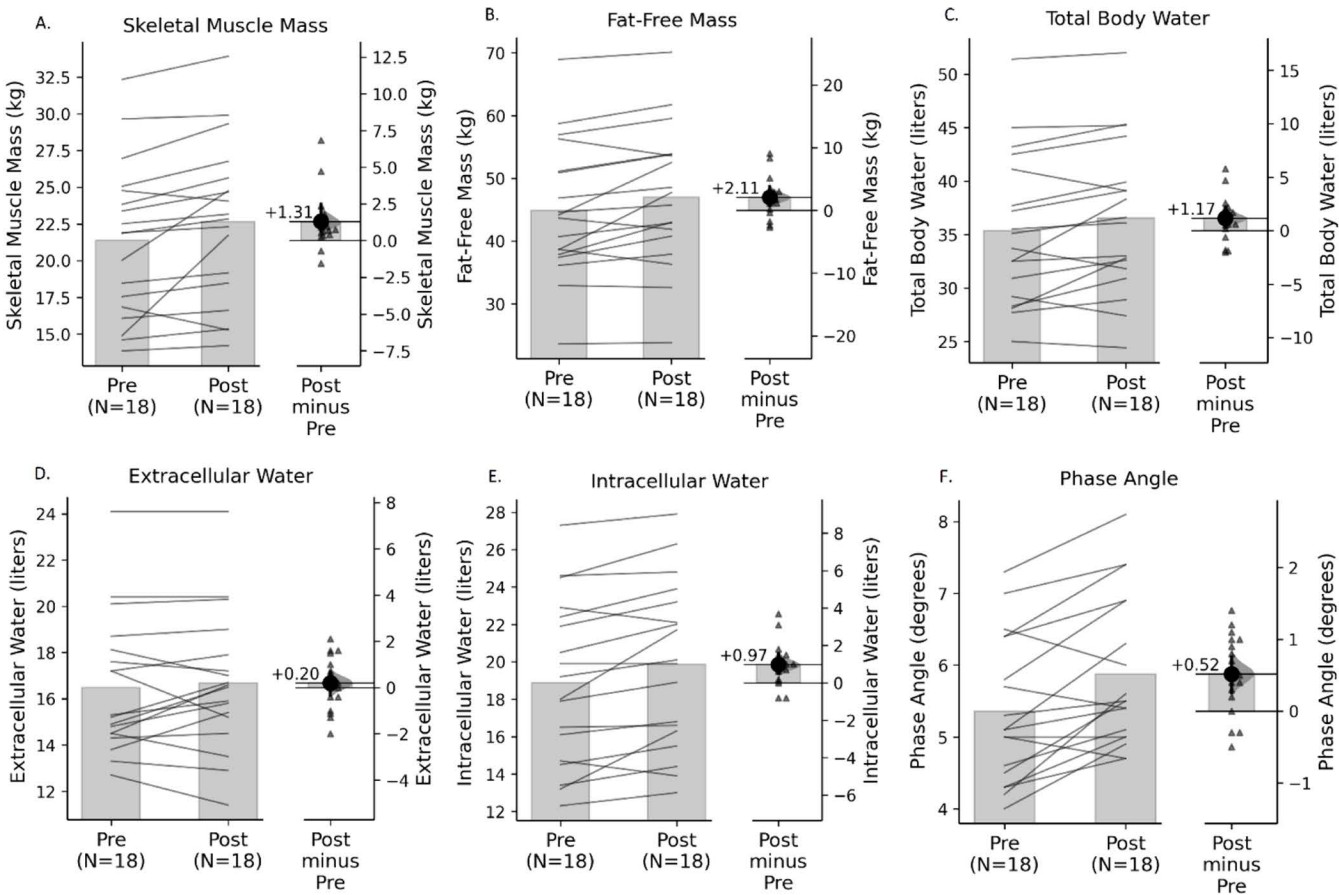

**Fig 2. Gardner-Altman plots showing changes from pre- to post-intervention. Markers of skeletal muscle mass are increased after creatine supplementation in HD patients.** Skeletal muscle mass (A), fat free mass (B) and phase angle (F) are increased in HD patients after receiving creatine supplementation for 8 weeks. Total (C) and intracellular water (E) also increased after creatine supplementation. Extracellular water (D) did not change after creatine supplementation. The left panel displays paired individual values, with lines connecting pre- and post-intervention measurements. The right panel illustrates the mean difference as a dot, with the 95% confidence interval indicated by the vertical bar. Positive values denote improvement after the intervention.

with an effect size of 0.81 indicating a large effect (Fig 2E). Extracellular water did not show significant changes before and after the intervention, with a mean difference of 0.20 ± 1.12 L [95% CI: −0.30 to 0.70], varying from 16.48 ± 2.96 L to 16.68 ± 2.98 L. The effect size was 0.17, indicating a negligible effect (Fig 2D).

In the stratified analysis by sex, significant differences in strength were observed among men, with an increase of 2.11 ± 2.96 kg [95% CI: 0.11 to 4.22]; the Hedges' g effect size was 0.57. Significant improvements were also found in SPPB scores in men, with an increase of 0.44 ± 0.76 points [95% CI: 0.11 to 0.89]; the Hedges' g effect size was 0.56 (Table 3).

Among female participants, a non-significant increase in SPPB scores was observed following the intervention, with a mean difference of 1.11 ± 1.8 [CI 95%: 0.00 to 2.22]. The calculated effect size (Hedges' g) was 0.55, suggesting a moderate magnitude of the observed effect. No improvement was observed in muscle strength in women following the intervention. The mean change was −0.77 ± 2.04 kgF [95% CI: −1.89 to 0.67]. The Hedges' g effect size was −0.35 (Table 3)

**Table 3. Bootstrap analysis of functionality tests and handgrip strength before and after creatine supplementation — subgroup analysis.**

|  | SPPB points [CI 95%][1] | Strength kgF [CI 95%][1] |
|---|---|---|
| Women (n:9) | 1.11 [0.00 to 2.22] | −0.77 [−1.89 to 0.67] |
| Men (n:9) | 0.44 [0.11 to 0.89]* | 2.11 [0.11 to 4.22]* |

[1]Point estimate and 95% confidence interval based on 10 000 bootstrap replicates of the pre–post differences.

*Statistically significant at 5% level.

## Discussion

In this study, we present the results of a pilot clinical trial demonstrating that functional capacity and body composition improved in hemodialysis patients supplemented with creatine. In line with our working hypothesis, creatine supplementation for eight weeks in individuals on chronic hemodialysis was associated with better scores in physical functionality tests (SPPB), along with favorable changes in body composition measured by bioimpedance, including increases in skeletal muscle mass, fat-free mass, and phase angle. These findings suggest that oral creatine supplementation may serve as a potential therapeutic strategy to enhance muscle mass and overall physical performance in hemodialysis patients.

For supplementation in the present study, we utilized creatine monohydrate, a supplement with extensive evidence supporting its safety [24–28], as well as its efficacy in improving muscular performance across various populations and clinical scenarios [29]. Creatine is a key substrate for energy metabolism in all tissues, particularly those with high ATP consumption, such as skeletal muscle, smooth muscle, cardiac muscle, brain, and neurons. Its mechanism of action involves an increase in phosphocreatine levels within muscle cells, optimizing the supply of phosphate groups for ATP generation [30]. Notably, creatine stores are primarily located in skeletal muscle cells, and oral supplementation can increase these reserves by approximately 25% [31]. Although our study did not measure intracellular creatine deposits, we observed significant changes in body composition and functionality which might be attributed to a hypothetical intracellular creatine increase.

In our study, we employed a continuous creatine dose of 5 grams per day, which is recommended for maintenance dose. This approach was chosen to avoid the potential risk of fluid overload associated with the loading dose strategy (20 g/day for seven days) [32], which requires the consumption of approximately one liter of water daily for creatine dilution. Our strategy is supported by a previous study in healthy volunteers demonstrating that a lower continuous dose over 28 days resulted in similar muscle creatine concentrations compared to a six-day loading regimen [33]. In the dialysis population, data on this issue are scarce. Therefore, we opted for an extended supplementation period, twice the duration of the referenced study, to ensure adequate replenishment considering the creatine losses occurring during each dialysis session.

The improvement in functionality, as measured by the SPPB test, is particularly noteworthy given the well-documented association between low physical functional capacity and mortality [34]. Following creatine supplementation, SPPB scores increased from a mean of 10.3 to 11.11 points (Fig 1B). Among the participants, only one patient exhibited a decline in SPPB score, while nine maintained their scores, and eight demonstrated improvement. The Hedges' g effect size of 0.53 indicates a moderate impact of creatine supplementation on SPPB performance. In the context of hemodialysis, where patients commonly experience progressive physical decline, even modest improvements may hold meaningful clinical relevance.

Although the subgroup analysis among female participants showed changes in SPPB test, with the lower bound of the confidence interval reaching 0.00, no definitive conclusions can be drawn given that this analysis included only 9 patients. While the observed effect size (Hedges' g) was moderate, the limited sample size substantially constrains the

robustness of these findings. The same caution applies to our observations in the male subgroup. Although the observed increase in muscle strength and SPPB in males appears potentially clinically meaningful—particularly considering the well-established association between muscle strength and mortality in dialysis patients—we consider it inappropriate to draw definitive positive conclusions at this stage. Nevertheless, as a pilot study, this work serves its purpose by generating preliminary evidence and contributing valuable insights that should inform future research with larger sample sizes and more rigorous designs, including the incorporation of a control group.

Supporting the relevance of our findings in relation to the improvements seen in the SPPB scores, a recent study by Uchida et al. [35] demonstrated that dialysis patients with lower SPPB scores have higher all-cause mortality, an increased risk of hospitalization, and greater hospitalization rates due to cardiovascular causes. Furthermore, for each point decrease in the SPPB score, patients were more likely to experience these adverse outcomes.

Given that SPPB is a well-established predictor of mobility and overall physical performance [36], our findings offer an interesting preliminary insight that may serve as a foundation for future studies with larger sample sizes and extended follow-up periods, aimed at determining whether the observed improvements in physical performance are sustained over time and whether they are associated with lower mortality rates.

Handgrip strength is a measure that deserves attention in order to its relationship to all cause mortality in hemodialysis patients [37]. In one of the few articles referring to creatine supplementation in dialysis patients, which evaluated changes in grip strength after one year of creatine plus neuromuscular electrical stimulation, significant increases in handgrip strength were observed [38]. Our findings show that after 8 weeks of creatine supplementation handgrip strength did not increase in the whole group. As mentioned earlier, the gender subgroup analysis showed significant increases in handgrip strength in men, but this observation has to be re-tested in future studies. As previous literature supporting this hypothesis came from patients in a different clinical context, the lack of a significant increase in handgrip strength in our whole group analysis could have been expected, as we supplemented for a shorter period and did not include an exercise or neuro-muscular electrical stimulation intervention as informed in Marini's work.

An additional noteworthy finding in our study, particularly in the bioimpedance analysis, was the increase in phase angle (Fig 2F). This parameter is recognized as an indicator of both the quantity and quality of soft tissues, with higher values reflecting greater cellularity, improved integrity, and enhanced cellular functionality [39]. Phase angle is calculated as the arc tangent of the Reactance/Resistance ratio, where resistance represents the opposition to the flow of electrical current through intra- and extracellular ionic solutions, and reactance reflects the opposition due to the capacitive properties of cell membranes and tissue interfaces [40]. Its value varies by gender and declines with age; in the general population, mean phase angle values range from 7° in men and 6.3° in women aged 39–48 years, decreasing to 5.1° in individuals aged 70–80 years [41]. In dialysis patients, lower phase angle values have been consistently associated with an increased risk of hospitalization, reduced physical function, poorer quality of life, and higher mortality [18,19]. In our study, we observed a statistically significant improvement of 0.52 degrees. The associated effect size of 0.90 suggests a large and potentially clinically meaningful change. However, no universally accepted threshold for phase angle variation has been established in the literature as a predictor of improved outcomes in dialysis patients. Nonetheless, a recent study proposed a cutoff value of 4.5°, suggesting that patients with phase angle values above this threshold may have lower mortality compared to those below it [20]. Given the established relationship between lower phase angle values and adverse outcomes, creatine supplementation could be a promising intervention for future long-term prospective studies to explore potential correlations between phase angle improvements and changes in quality of life and and even in terms of mortality in dialysis patients.

Other variables in which we observed significant differences include fat-free mass and skeletal muscle mass, both of which increased following creatine supplementation. Considering that skeletal muscle mass is a key component in the diagnosis of sarcopenia and protein-energy wasting [42,43]—conditions with high prevalence among hemodialysis patients and a strong association with mortality [44–47]—these findings appear desirable in this population. However,

studies with longer follow-up periods and more rigorous designs are needed to establish any relationship with clinically relevant outcomes, such as mortality or hospitalization.

When these findings are considered alongside changes in body water compartments, it is reasonable to infer that our results are likely related to an increase in ICW. It is well known that creatine promotes intracellular water retention, seemingly driven by its co-transport with sodium into the intracellular space [48]. While this finding might seem unremarkable, it could have important implications given a recently published study showing that dialysis patients with low baseline post-dialysis ICW correlated with muscle wasting and inflammation and was an independent risk factor for mortality [49]. It is also important to note that evidence in humans has demonstrated increased muscle protein synthesis, greater muscle glycogen content, reduced markers of protein catabolism, and an increased number of satellite cells following creatine supplementation [50,51]. Therefore, it could be speculated that, as observed in other studied populations, the increase in muscle mass may not be solely attributable to changes in intracellular hydration, but also to these additional anabolic mechanisms. Nevertheless, histological and molecular biology studies are needed to confirm these effects in the dialysis population.

Importantly, although both TBW and ICW increased significantly following creatine supplementation, no significant change was observed in dry weight. This likely reflects the relatively small magnitude of the TBW increase (1.17 liters), which falls well within the known variability of dry weight measurements in dialysis patients. Previous studies have shown that fluctuations of 1–2 kg in dry weight between sessions are common and do not necessarily indicate fluid overload [52]. Although the change in total body water reached statistical significance, its magnitude is unlikely to translate into a clinically noticeable alteration in dry weight, likely due to the inherent variability in fluid distribution and limitations in clinical assessment. Notably, ECW remained stable throughout the intervention, which supports the safety and tolerability of creatine supplementation in this population. It is also important to consider that in patients undergoing thrice-weekly chronic hemodialysis—particularly those who are anuric, comprising 50% of our study population—extracellular water is routinely regulated through ultrafiltration during each session. This therapeutic process, aimed at removing excess fluid alongside solute clearance, likely contributed to maintaining stable dry weight despite the observed shifts in TBW.

To date, only one study has examined body water compartments in dialysis patients receiving creatine supplementation, and it reported findings similar to ours in this regard. However, unlike our study, it did not assess physical functional outcomes [53].

Laboratory values, including hematocrit, calcium, phosphorus, blood urea nitrogen, albumin, nPCR, Kt/V, and dry weight, remained unchanged before and after supplementation. It is important to note that both Kt/V and nPCR are routinely applied in dialysis units worldwide and are derived from objective, patient-specific clinical parameters [54,55]. Importantly, these formulas do not incorporate race or ethnicity as variables. Therefore, we do not consider that their use introduced any bias in the population included in our study.

We did observe a significant increase in serum creatinine, which was an expected result given the metabolism of creatine to creatinine [56]. Additionally, vitamin D levels showed a significant increase post-supplementation. Upon analysis, two primary factors were identified as potential contributors: some patients (n = 8) received both creatine and vitamin D supplementation due to initially low plasma vitamin D levels, and seasonal variation, with pre- and post-supplementation measurements taken in winter and summer, respectively, a phenomenon known to influence vitamin D levels [57]. Since vitamin D is linked to functionality and strength in dialysis patients [58], we further analyzed functionality outcomes only in patients with increased vitamin D levels (n = 14) and found no significant difference in SPPB scores (S1 Table), ruling out confounding effects.

Consistent with previous studies using creatine in patients undergoing renal replacement therapy [59,60], no adverse effects were reported among our study participants. No patient experienced diarrhea, abdominal discomfort, or any other symptoms that required discontinuation of its use.

Among the strengths of our pilot study is its status as the first to evaluate the impact of creatine supplementation on functional levels in patients with end-stage renal disease in dialysis. However, we acknowledge certain limitations, including the small sample size (n = 18), the absence of a control group to account for the placebo effect and that we did not reach the calculated sample size; however, we attempted to address this limitation by conducting a robust statistical analysis of our findings, which yielded results showing statistical significance. Another limitation of this study is the use of convenience sampling, which, while practical in the clinical setting, may have introduced selection bias. This approach could limit the external validity of our findings, as the sample may not be fully representative of the general dialysis population. However, it is important to note that the selected group had a similar average age to that of the overall patient population in our unit, and that gender equity was achieved in the sample. Finally, although there are more precise methods for measuring muscle mass, such dual energy X-ray absorptiometry (DEXA) [61] and isotope dilution techniques for the assessment of body water compartments [62], they are unfortunately not currently accessible in our region. Therefore, we relied on bioelectrical impedance analysis (BIA), given its significantly greater accessibility and widespread clinical use worldwide. However, it would be of interest to confirm these findings using those reference techniques in settings where such technology is available. Future research involving a larger cohort and a control group is necessary to establish a definitive relationship between creatine supplementation and functional outcomes in dialysis patients.

While this study provides valuable preliminary insights into creatine supplementation in dialysis patients, several uncontrolled variables may have influenced the results. Dietary variations, particularly protein intake and consumption of creatine-rich foods, were not recorded during the study and could have contributed to baseline differences between participants. Also individual physical activity levels that naturally vary among individuals were not registered, representing another potential confounding factor. Additionally, seasonal variations during summer months may have influenced hydration status and vitamin D levels. With these possible confounders in mind, throughout the course of the trial, all patients received nutritional counseling provided by the same dietitian. They were also told to keep their baseline, regular physical activity levels during the study. Although these uncontrolled factors introduce some uncertainty in attributing outcomes only to creatine supplementation, they reflect real-world conditions and do not necessarily invalidate the findings, but rather highlight areas for standardization in future controlled trials.

The most relevant finding of this study is the observed improvement in SPPB test scores following creatine supplementation in dialysis patients, accompanied by favorable changes in body composition. Considering the established association between reduced physical function and adverse outcomes, these preliminary findings suggest that creatine supplementation may represent a promising strategy to improve functional status in this high-risk population. However, larger and adequately powered studies are warranted to confirm these results and to evaluate potential effects on clinical outcomes such as mortality and hospitalization.

## Supporting information

**S1 Table. SPPB score pre and post creatine supplementation in patients with increased vitamin D levels.** Mean difference (post – pre): 0.85 [95% CI: 0.00–1.54], based on 10,000 bootstrap replicates of the paired differences. (DOCX)

**S1 File. CONSORT 2025 editable checklist.**
(DOCX)

## Acknowledgments

We would like to express our sincere gratitude to the nursing team of the dialysis unit at Hospital Puerto Aysén.

## Author contributions

**Conceptualization:** Waldo Bernales-Delmon, Iván Guglielmi, María Claudia Martínez, Carla Basualto-Alarcón.

**Data curation:** Waldo Bernales-Delmon, Juan John, Felipe Matus.

**Formal analysis:** Waldo Bernales-Delmon, Simón Schulz, Carla Basualto-Alarcón.

**Funding acquisition:** Iván Guglielmi, Carla Basualto-Alarcón.

**Investigation:** Waldo Bernales-Delmon, Jaime Joost, José Aguilar, Andrés Wulf, Paulina Bittner, María Claudia Martínez, Sandy Gómez, Catalina Chávez, Juan John, Felipe Matus, Carla Basualto-Alarcón.

**Methodology:** Waldo Bernales-Delmon, Simón Schulz, Carla Basualto-Alarcón.

**Project administration:** Yasna Venegas, Iván Guglielmi, Cynthia Saravia, Carla Basualto-Alarcón.

**Resources:** Waldo Bernales-Delmon, Yasna Venegas, Cynthia Saravia.

**Software:** Simón Schulz.

**Supervision:** Waldo Bernales-Delmon, María Claudia Martínez, Sandy Gómez.

**Validation:** Waldo Bernales-Delmon, Simón Schulz, Carla Basualto-Alarcón.

**Visualization:** Waldo Bernales-Delmon, Carla Basualto-Alarcón.

**Writing – original draft:** Waldo Bernales-Delmon, Simón Schulz, Carla Basualto-Alarcón.

**Writing – review & editing:** Waldo Bernales-Delmon, Simón Schulz, Yasna Venegas, Iván Guglielmi, Cynthia Saravia, Jaime Joost, José Aguilar, Andrés Wulf, Paulina Bittner, María Claudia Martínez, Sandy Gómez, Catalina Chávez, Juan John, Felipe Matus, Carla Basualto-Alarcón.

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
