## [Decision Letter · Decision Letter 0]

Dear Dr. Basualto-Alarcón,

Thank you for submitting your manuscript to PLOS ONE. After careful consideration, we feel that it has merit but does not fully meet PLOS ONE’s publication criteria as it currently stands. Therefore, we invite you to submit a revised version of the manuscript that addresses the points raised during the review process.

MAJOR DETAILS:

1. The structure of the manuscript, especially the METHODS section, needs to be revised. All subsections should be organized in strict order as recommended by CONSORT guidelines for pilot and feasibility studies (Trial design, Participants, Intervention Procedures, Outcomes, Sample Size, Statistical Analysis) as informed in the submission guidelines of PLoS One. Cf, PMID 31608150 or www.equator-network.org.

Was the clinical trial registered? Platforms such as ClinicalTrials.gov, OSF, or Figshare are available for this purpose.

Were participants aware that they were receiving the creatine monohydrate supplement (i.e., open-label study), or were they blinded to the intervention (i.e., single-blind)? Consider including this information in the title for greater clarity.

This study is not a prospective observational study. This is a single-arm repeated-measures feasibility clinical trial (revise Line 126 and line 300). This is an experimental study.

The authors should clarify which variables are considered primary outcomes and which are secondary outcomes.

Was dietary intake or physical activity recorded during the study?

Please provide additional details regarding the blood sampling procedures, the test protocols used to evaluate biomarkers, and the equipment or devices employed in the laboratory.

Have the formulas been validated in the study population, or at least in a Latin American population? If not, this should be acknowledged as a limitation. 

2. Replace conventional bioelectrical impedance analysis with bioelectrical impedance vector analysis (BIVA). If you used the mBC 565 device, you should present BIVA results—such as the Xc versus R plot normalized by stature, or even specific BIVA using available body girth measurements—instead of relying solely on body composition estimates of SMM, ICW, FFM, ECW, and TBW. While phase angle is a valuable absolute outcome of BIVA, it can be further complemented with indices like the BI index (Height²/Z at 50 kHz) and the impedance ratio (Z at high and low frequencies in kHz).

"An IR ratio closer to 1 is indicative of cell membrane disruption, allowing more fluids, proteins, and electrolytes to shift into the extracellular space [22]. A strong inverse correlation has been reported between the phase angle (PhA) and IR in different clinical populations [22]. PhA is defined as the delay in current flow caused by a reduction in cell membrane capacitance [25]." Cf, PMID: 38488531

3. Statistical analysis. 

This section requires revision. Due to the small sample size and the fact that the required statistical power was not achieved (i.e., fewer participants than indicated by the sample size calculation), the authors should avoid null hypothesis significance testing (NHST). Instead, the analysis should focus on estimation methods—reporting 95% confidence intervals and unbiased effect sizes (e.g., Cohen's d, also known as Hedges' g)—as well as robust statistics, such as trimmed means and Winsorized standard deviations. Please re-run the analysis accordingly and confirm whether the findings and conclusions remain consistent. Thanks to your open data sharing, I have attached the raw data Excel file along with an example of the results output from Jamovi, which you may use for organizing and presenting your findings.

Cf,

- Estimation statistics: https://pubmed.ncbi.nlm.nih.gov/24220629/

"ESCI" module in Jamovi

- Robust statistics: https://pubmed.ncbi.nlm.nih.gov/31152384/

"Walrus" module in Jamovi

In addition, please generate and replace current figures with estimation plots to display the repeated measures data across time points (at baseline and after creatine supplementation). Cumming or Gardner-Altman estimation plots are recommended (see examples of these figures in the attached Excel file).

Please add the CV% of the instruments/technician to know the variability of all measures.

MINOR DETAILS: 

- Please adjust the background and hypothesis in the introduction secion (as well as in the abstract) to creatine related information. The introduction can be reduced to one page only. Missing key references on this topic: 

https://pubmed.ncbi.nlm.nih.gov/31083291/

https://pubmed.ncbi.nlm.nih.gov/28110688/

https://pubmed.ncbi.nlm.nih.gov/34444869/

- Although "body weight" and "height" are frequently used terms, it is technically correct to refer to "body mass" and "stature", respectively. Please address this accordingly throughout the manuscript.

- Dynamometer strength measurements should be expressed in kilogram-force (kgF). Please revise accordingly and replace "Strength" with "Handgrip Strength" in Figure 1. 

- Do not use "mean±standard deviation." Use Mean(SD) instead. Cf, PMID 21206631

- The results of all variables should be expressed as Δ ± SD [95% CI of the change]; unbiased Cohen's [with the corresponding 95% CI]. Remember that unbiased Cohen's = Hedge's g

- Please edit the tables to present all relevant findings, including effect sizes, 95% confidence intervals, and other pertinent statistics.

- Line 374: The authors should highlight that this is one of the first clinical trials to evaluate the effects of oral creatine monohydrate supplementation on BIVA variables, strength-related outcomes, and functional capacity in hemodialysis patients. 

- The discussion should place greater emphasis on BIVA-related data (e.g., bioimpedance indices reflecting cellular integrity), strength-related outcomes, and estimates of body composition (e.g., skeletal muscle mass, fat-free mass, etc.).

- The discussion on intra- and extra-celullar water might be enriched if the impedance ratio is reported (see comments above).

- In agreement with the reviewers, the authors should elaborate on potential confounding variables (e.g., diet, physical activity, season time, etc.) that could have influenced or biased the findings. 

- Additional information that should be referenced in the discussion to support the safety of creatine supplementation on renal health includes: 

https://pubmed.ncbi.nlm.nih.gov/31375416/

https://pubmed.ncbi.nlm.nih.gov/38869518/

https://pubmed.ncbi.nlm.nih.gov/38874125/ 

- Please provide a short paragraph of limitations before conclusions.

- Acknowledge conflicts of interest (if any) in the acknowledgements section.

- Replace the "STROBE checklist" with "CONSORT for feasibility and pilot studies checklist"

We look forward to receiving your revised manuscript.

Kind regards,

**Prof. Diego A. Bonilla**

Academic Editor

PLOS ONE

Additional Editor Comments:

Dear authors,

The manuscript "Oral creatine in hemodialysis patients increases physical functional capacity and muscle mass" is interesting and could be a valuable contribution to the literature.

This clinical study aimed to determine whether oral creatine monohydrate supplementation improves muscle mass and physical functional performance in a convenience sample of adult dialysis patients, with measurements taken before and after supplementation. However, the authors need to address the reviewers' concerns and revise the manuscript in accordance with the editor's comments to improve its structure and data analysis, thereby enhancing its scientific robustness. I commend the authors for supporting open science and sharing their data.

Reviewers' comments:

Reviewer's Responses to Questions

**Comments to the Author**

1. Is the manuscript technically sound, and do the data support the conclusions?

Reviewer #1: Yes

Reviewer #2: Partly

Reviewer #3: Yes

2. Has the statistical analysis been performed appropriately and rigorously?

Reviewer #1: Yes

Reviewer #2: Yes

Reviewer #3: Yes

3. Have the authors made all data underlying the findings in their manuscript fully available?

Reviewer #1: Yes

Reviewer #2: No

Reviewer #3: Yes

4. Is the manuscript presented in an intelligible fashion and written in standard English?

Reviewer #1: Yes

Reviewer #2: Yes

Reviewer #3: Yes

Reviewer #1: Peer Review: "Oral Creatine in Hemodialysis Patients Increases Physical Functional Capacity and Muscle Mass"

Below is my peer review of the manuscript entitled "Oral Creatine in Hemodialysis Patients Increases Physical Functional Capacity and Muscle Mass." This study is very interesting and timely. However, the authors need to address some of my concerns prior to acceptance for publication.

Major Concerns:

1. Absence of a Control Group

• Description: The study employs a single-arm, before-and-after design without a control group. All 18 participants received creatine supplementation (5 g/day for 8 weeks), and outcomes were compared pre- and post-intervention without a comparator group receiving placebo or standard care.

• Reference:

• Methods section, "Patient Selection" (Page 14, Lines 124-137): "All the recruited patients received creatine supplementation. They were measured before (pre-) and after (post-) receiving creatine supplementation."

• Introduction (Page 13, Line 118): It is an observational study.

• Impact: The lack of a control group is a significant methodological flaw that compromises the study's internal validity. Without a control, it is impossible to attribute the observed improvements in skeletal muscle mass, intracellular water, phase angle, and Short Physical Performance Battery (SPPB) scores solely to creatine supplementation. Confounding factors such as the placebo effect, regression to the mean, natural recovery, seasonal variations (e.g., vitamin D levels increased from winter to summer, Page 21, Lines 241-245), or increased attention from researchers could explain the results. This limitation undermines the study's ability to establish causality, a critical aspect for a research article claiming that creatine "increases" functional capacity and muscle mass, as stated in the title and conclusion (Page 11, Lines 60-63).

• Significance: For a pilot study, a single-arm design may be exploratory, but the absence of a control group severely limits the strength of the conclusions, especially given the definitive language used (e.g., "improved muscular and functional outcomes," Page 11, Line 61).

• Recommendation: Revise the manuscript to explicitly acknowledge the lack of a control group as a significant limitation in the Discussion section (beyond the brief mention on Page 28, Line 400). Temper the conclusions to reflect the preliminary nature of the findings, e.g., change "increases physical functional capacity and muscle mass" (Title, Page 1) to "may increase" or "is associated with increases in." Suggest a randomized controlled trial (RCT) as the next step to confirm causality.

• Rationale: This adjustment aligns the interpretation with the study's observational design and mitigates overstatement, preserving credibility.

2. Insufficient Sample Size

• Description: The study calculated a required sample size of 25 patients based on an effect size of 0.6, alpha of 0.05, and power of 0.8 for detecting changes in SPPB scores (Page 17, Lines 182-190). However, only 19 patients were recruited, and 18 completed the study, falling short of the planned sample size.

• Reference:

• Methods section, "Sample size calculation" (Page 17, Lines 182-190): "The required sample size... is 25 patients... we were only able to approach the calculated sample size."

• Results section, "Baseline Characteristics" (Page 19, Lines 210-216): "We were initially able to recruit only 19... From this initial selection, we had to withdraw one patient."

• Impact: Failing to meet the calculated sample size raises concerns about the study's statistical power. Although significant results were reported (e.g., SPPB p=0.043, skeletal muscle mass p=0.009, Page 22, Table 2), an underpowered study increases the risk of type II errors (missing true effects) and may inflate the risk of type I errors (false positives), particularly with borderline p-values like 0.043 for SPPB.

• Significance: The small sample size, combined with the lack of a control group, amplifies doubts about the robustness of the conclusions. The authors acknowledge this limitation (Page 28, Line 399), but the shortfall still constitutes a significant concern given the study's reliance on statistical significance to support its claims.

• Recommendation: Expand the Discussion to detail the implications of the reduced sample size on statistical power and result reliability (beyond the current note on Page 28, Line 399). Report post-hoc power analyses for key outcomes (e.g., SPPB, skeletal muscle mass) to clarify the strength of the findings. Recommend future studies with adequate sample sizes based on the observed effect sizes (e.g., 0.53 for SPPB, Page 25, Line 335).

• Rationale: Transparency about power limitations strengthens the manuscript's integrity and guides future research design.

3. Selection Bias Due to Convenience Sampling

• Description: The study used a convenience sample of patients selected based on "good adherence to pharmacological and dialysis therapy" (Page 14, Lines 125-126). This non-random selection method resulted in a cohort that may not represent the broader hemodialysis population.

• Reference:

• Methods section, "Patient Selection" (Page 14, Lines 124-131): "A convenience sampling... primarily based on their good adherence to pharmacological and dialysis therapy."

• Results section, "Baseline Characteristics" (Page 19, Lines 210-216): Describes recruitment challenges, noting exclusions due to mobility issues and poor adherence.

• Impact: Selecting patients with good adherence introduces selection bias, as these individuals may be healthier, more compliant, or more motivated than the average hemodialysis patient. This bias limits the external validity of the findings, as the results may not generalize to the broader population, particularly those with poorer adherence or more severe conditions who might respond differently to creatine supplementation. The manuscript does not adequately address how this selection impacts the applicability of the findings to clinical practice.

• Significance: This methodological flaw questions the study's relevance to the target population stated in the introduction (Page 13, Lines 88-93), where hemodialysis patients are described as a vulnerable group with high morbidity and mortality. The biased sample weakens the claim that creatine could mitigate these outcomes across all hemodialysis patients (Page 11, Lines 61-63).

• Recommendation: Discuss the potential impact of selection bias in the Discussion section, noting that patients with good adherence may differ from the general hemodialysis population in health status or behavior. Adjust claims about generalizability (e.g., Page 11, Lines 61-63) to specify the study population (e.g., "in adherent hemodialysis patients"). Propose stratified sampling or broader inclusion criteria in future studies.

• Rationale: Acknowledging and contextualizing this bias enhances the manuscript's applicability and prevents overgeneralization.

Conclusion

The manuscript presents valuable preliminary data suggesting that creatine supplementation may benefit hemodialysis patients by improving muscle mass and functional capacity. However, the identified significant concerns—lack of a control group, insufficient sample size, and selection bias—fundamentally undermine the study's ability to establish causality, reliability, and generalizability. These issues do not invalidate the research entirely, especially given its pilot study context, but they require substantial revision to ensure scientific soundness.

Additional Recommendation for Future Research

Consider a double-blind RCT with a placebo control group, a more prominent and more representative sample, and monitoring of confounders (e.g., physical activity, diet) to address all three concerns comprehensively.

Reviewer #2: This study evaluated the effect of 8 weeks of daily creatine monohydrate supplementation in patients with end stage CRF on dialysis.

The study reported improvements in SPPB and ICW with an increase in FFM and muscle mass.

Comments to consider:

1. The study was open label and there was no control group. As the authors indicated, most subjects started in the winter and finished in the summer. This was likely one contributor to the higher vitamin D (in addition to the 8 who were supplemented) BUT was also likely associated with more physical activity. Most healthy people and patients tend to do more physical activity in the summer vs winter and this could confound the SPPB scores. The trend for those who had increased (I am assuming that the levels increased over time and not increased above normal) was rather strong (P = 0.085) suggesting that vitamin D could be a factor (keep in mind that the sample size was small for the sub-group).

2. I do not see that the authors instructed the patients to control for diet, hydration status and pre-measurement exercise that can confound the BIA measurements.

3. The lack of change in body weight and a ~ 2 kg increase in FFM implies that body fat was down ~ 2 kg - the authors need to comment on this and provide some explanation.

4. The authors should discuss the limitations of the BIA and the lack of another method for the analysis such as DEXA.

Reviewer #3: This is a very interesting study. The hypothesis is well founded and based on relevant literature and the methods and results are sound.

In my opinion, this study should certainly be accepted for publication, provided some minor revisions are made.

The primarily limitation is the open label nature of the study, but this issue can be resolved by sufficiently stating this.

Comments:

Comment 1: Please include a short mention in the methods what an increase in phase angle represents.

Comment 2: The open label nature of the study should be reflected in the abstract. Please include the words “open label” or state there was no control group.

Comment 3: The term protein-energy-wasting is not mentioned in the introduction or discussion, please incorporate this too.

Comment 4: “Different nutritional supplements have also been explored, However, its implementation is often associated with higher costs and the need for trained personnel to ensure its proper operation.”, please elaborate which supplements are referred to with this statement, or omit alltogether.

Comment 5: The quality of the figures appears to be quite low. Please improve to 300 dpi.

Comment 6: How was intracellular water calculated? Please add.

Comment 7: How was missing data handled in the analyses?

Comment 8: Please define how dry weight was determined.

Comment 9: Please report in SI units, i.e. μmol/L rather than mg/dL.

Comment 10: There is no mentioning of side effects in the results section (only shortly in the discussion). Please report on this in the results section too. Did patients experience for instance bloating/nausea or interdialytic weight gain due to the extra volume?

Comment 12: Do you have any explanation why no effect was seen on handgrip strenght, although this was hypothesized? Please discuss this.

Comment 13: Line 86, omit additional space; Line 100, double end of sentence point after [8-9]

**Do you want your identity to be public for this peer review?** For information about this choice, including consent withdrawal, please see our Privacy Policy

Reviewer #1: No

Reviewer #2: No

Reviewer #3: No

---

## [Author Response · Author response to Decision Letter 1]

25 Jun 2025

Rebuttal letter

Coyhaique, June 20th, 2025

Dear editor and reviewers,

We would like to thank you for the thoughtful and constructive review of our manuscript. The observations and suggestions provided have certainly prompted us to improve and enrich our paper, making it more rigorous and scientifically robust. In the next paragraphs you will find our responses in the same order as we received the observations and answered point by point. We hope we have been able to respond clearly to each of your comments and suggestions.

MAJOR DETAILS:

1. The structure of the manuscript, especially the METHODS section, needs to be revised. All subsections should be organized in strict order as recommended by CONSORT guidelines for pilot and feasibility studies (Trial design, Participants, Intervention Procedures, Outcomes, Sample Size, Statistical Analysis) as informed in the submission guidelines of PLoS One. Cf, PMID 31608150 or www.equator-network.org.

Answer 1: We thank the editor for raising this point.

During the process of designing our study we consulted the NIH web page: Decision Tool: Does Your Human Subjects Study Meet the NIH Definition of a Clinical Trial? (https://grants.nih.gov/policy-and-compliance/policy-topics/clinical-trials/ct-decision). After careful consideration we thought that our study didn’t meet the “four questions to determine if your study is a clinical trial”, and for this reason we decided to report it as an observational study, following the STROBE guidelines. Although, we agree with the editor, that this study can be also considered an experimental study (a “single-arm repeated-measures feasibility clinical trial”). The revised version of our paper has changed the methods sections and now presents the subsections organized as recommended by CONSORT guidelines (please see CONSORT checklist).

Was the clinical trial registered? Platforms such as ClinicalTrials.gov, OSF, or Figshare are available for this purpose.

Answer 2: Since we did not consider our study to be a clinical trial, it was not enrolled in any of the respective platforms, before starting patient recruitment. We are currently in the process of enrolling our study a posteriori, on clinicaltrials.gov platform (Universidad de Aysén Protocol Record ORD252023, in process). We also reviewed the other two alternatives proposed by the editor (Figshare and OSF), however, these appear to be platforms for sharing data and projects during their development, but did not have the features shared by the platforms suggested by the World Health Organization, grouped in the International Clinical Trials Registry Platform (ICTRP).

As our institution was not previously enrolled in clinicaltrials.gov platform, the process for the institution to be registered and the posterior steps took more time that the one considered for answering the reviewers comments. For this reason, the clinicaltrials.gov enrollment process is still ongoing. We understand this information is important and still pending. As soon as the registration is complete, we will send the information.

Were participants aware that they were receiving the creatine monohydrate supplement (i.e., open-label study), or were they blinded to the intervention (i.e., single-blind)? Consider including this information in the title for greater clarity.

Answer 3: All participants were aware that they were receiving creatine. We included the phrase 'open-label' in the title to clarify this point, and also mentioned it in the creatine supplementation section of the Methods (lines 123-124, clean version).

This study is not a prospective observational study. This is a single-arm repeated-measures feasibility clinical trial (revise Line 126 and line 300). This is an experimental study.

Answer 4: We agree with the editor. Please see our more complete response to this issue, in the comment above (“answer 1”). We have made the changes accordingly.

The authors should clarify which variables are considered primary outcomes and which are secondary outcomes.

Answer 5: We thank the editor for highlighting this important point and allowing us to clarify these aspects.

The primary outcome of this study was to evaluate the physical performance of hemodialysis patients before and after receiving creatine supplementation as assessed by the Short Physical Performance Battery (SPPB).

The secondary outcomes in the current study were to demonstrate changes in muscle strength, measured by hand dynamometry, and changes in body composition, estimated by bioimpedance analysis (skeletal muscle mass, fat free mass, total body water, extracellular water and intracellular water). Additionally, we aimed to evaluate changes in phase angle before and after eight weeks of creatine supplementation.

This information has been explicitly included in the Materials and Methods section, (lines 125-135, clean version).

Was dietary intake or physical activity recorded during the study?

Answer 6: This is a very important point and we, deliberately, decided not to modify or register dietary intake nor physical activity. The former decision was taken based on bibliography that demonstrates that self-monitoring has significant impacts on the measured variables inducing changes on dietary intake and physical activity (Burke LE, Wang J, Sevick MA. Self-monitoring in weight loss: a systematic review of the literature. J Am Diet Assoc., 2011: 111(1):92-102. doi: 10.1016/j.jada.2010.10.008; Tang, HB., Jalil, N.I.A., Tan, CS. et al. Why more successful? An analysis of participants’ self-monitoring data in an online weight loss intervention. BMC Public Health, 2024: 24, 322. doi.org/10.1186/s12889-024-17848-9).

We also did not recommend any changes to the participants' usual diet; instead, we maintained the general dietary recommendations routinely provided by the dietitian at our dialysis unit, primarily focusing on foods low in phosphate, sodium, and potassium (lines 239-241, clean version).

Regarding physical activity, it was not recorded in this first pilot study. We assessed physical performance exclusively using the Short Physical Performance Battery (SPPB). However, we are currently seeking funding to conduct a new study—a randomized, double-blind, multicenter trial with a larger patient population—in which we will include physical activity assessment using the International Physical Activity Questionnaire (IPAQ).

Please provide additional details regarding the blood sampling procedures, the test protocols used to evaluate biomarkers, and the equipment or devices employed in the laboratory.

Answer 7: The national regulations related to the care of patients in chronic hemodialysis, consider the evaluation of specific routine laboratory tests. Some of these tests are performed monthly, while others are carried out quarterly or semiannually. As in many other countries, it is standard practice to collect monthly blood samples prior to dialysis during the second session of the week, which typically falls on Wednesday or Thursday, depending on the patient's schedule. Additionally, post-dialysis blood samples are taken at the end of the same session, primarily to calculate the dialysis dose using widely accepted and standardized formulas such as Kt/V.

The blood samples collected in the dialysis unit are subsequently transferred to the hospital laboratory, which is located in the same building. Biochemical analyses—including electrolytes, calcium, parathyroid hormone (PTH), and vitamin D—are performed using the ARCHITECT ci4100 analyzer (Abbott). Complete blood counts are analyzed using the CELL-DYN Ruby system (Abbott), (lines 149-150, clean version).

Have the formulas been validated in the study population, or at least in a Latin American population? If not, this should be acknowledged as a limitation.

Answer 8: We thank the editor for pointing this out, as we had mistakenly assumed it was a general topic when it is actually part of the specialized technical knowledge handled primarily by professionals working in dialysis units every day. The following is a brief description of the history and use of each of these parameters. For the sake of clarity in the text, we have incorporated a clarifying paragraph and included the article (lines 468 - 472, clean version).

The Kt/V and nPCR formulas are routinely used in dialysis units worldwide. These formulas have been extensively studied and analyzed since the 1980s, when efforts began to establish methods for quantifying dialysis dose and correlating it with patient mortality. Although, to our knowledge, they have not been specifically validated in Latin American populations, they are universally accepted without distinctions based on race or gender. The Kt/V formulas, such as those proposed by Daugirdas are based on objective, patient-specific parameters—including pre- and post-dialysis urea levels, dialysis duration, ultrafiltration volume, and post-dialysis body weight—and do not incorporate race or ethnicity as a variable. Similarly, the nPCR formula, which is used to assess protein catabolism in dialysis patients, also relies on objective measures, such as urea concentrations and body weight, and does not include any race-based adjustments. Unlike certain estimations of glomerular filtration rate (e.g., Modification of Diet in Renal Disease, MDRD), which historically included race-based adjustments, both Kt/V and nPCR are applied uniformly across populations. Therefore, the application of both Kt/V and nPCR in our study population is consistent with international clinical practice.

Based on the aforementioned, we do not believe that the data obtained using these formulas could introduce any bias in the population included in our study.

2. Replace conventional bioelectrical impedance analysis with bioelectrical impedance vector analysis (BIVA). If you used the mBC 565 device, you should present BIVA results—such as the Xc versus R plot normalized by stature, or even specific BIVA using available body girth measurements—instead of relying solely on body composition estimates of SMM, ICW, FFM, ECW, and TBW. While phase angle is a valuable absolute outcome of BIVA, it can be further complemented with indices like the BI index (Height²/Z at 50 kHz) and the impedance ratio (Z at high and low frequencies in kHz).

"An IR ratio closer to 1 is indicative of cell membrane disruption, allowing more fluids, proteins, and electrolytes to shift into the extracellular space [22]. A strong inverse correlation has been reported between the phase angle (PhA) and IR in different clinical populations [22]. PhA is defined as the delay in current flow caused by a reduction in cell membrane capacitance [25]." Cf, PMID: 38488531

Answer 9: Thank you for your valuable comment regarding the inclusion of additional bioimpedance-derived parameters.

Regarding this point, and taking into account new information we obtained while addressing another of your suggestions—specifically the request to provide the coefficient of variation (CV%) of the bioimpedance device used—we have decided to modify the focus of our discussion.

Our search revealed that the only reported coefficient of variation (CV%) for the device refers to skeletal muscle mass, with values of 0.4% for intra-operator and 0.6% for inter-operator assessments. For estimated variables such as total body water (TBW) and extracellular water (ECW), studies in South American populations report a standard error of estimate (SEE) of 1.4 liters for TBW and 0.7 liters for ECW. (seca GmbH & Co. KG. Instructions for use: seca mBCA 525 / seca mBCA 535. Hamburg (Germany): seca GmbH; 2022. p. 126. Available from: https://www.seca.com/).

Therefore, and following your suggestion, we considered it was more appropriate to focus our analysis on variables that are either directly measured or supported by stronger evidence and smaller coefficients of variation. We also considered it important that the information derived from this and future studies be as clear and interpretable as possible for the multidisciplinary teams involved in dialysis care—as well as for the patients themselves. Accordingly, we are now proposing to center our analysis primarily on a variable that is directly calculated by the bioimpedance device—without requiring additional software or complex formulae—thus facilitating both data acquisition and clinical interpretation. This variable is phase angle, which is derived from raw measurements (resistance and reactance) and is provided directly in the device’s standard report, as is also the case with other commercial bioimpedance analyzers, without the need for further manual data processing.

As a second approach, we propose focusing on skeletal muscle mass, as its estimation via bioimpedance is supported by validated cut-off values and has been explored in prior studies—particularly in the context of creatine supplementation in dialysis patients (Marini, et al, see reference number 53). As a secondary analysis, we will describe the changes observed in body water compartments (TBW, ECW, and intracellular water [ICW]), acknowledging that these are less precise estimations.

Based on your suggestions we conducted a re-analysis, exploring additional indices such as the bioimpedance index or impedance ratio. Although valuable as a more direct representation of bioimpedance measurements, we found it difficult for those concepts to have a generalized clinical applicability. These metrics introduce concepts that may be difficult for healthcare teams and patients to interpret and, additionally, require a more exhaustive data extraction and processing effort by the clinical team.

Nevertheless, we conducted bootstrap analyses of the differences in resistance and reactance, and also calculated changes in the bioimpedance index using the same statistical approach. These results are attached in the following table, should you consider them appropriate for inclusion in the final manuscript version (please see table at the Word version uploaded).

3. Statistical analysis.

This section requires revision. Due to the small sample size and the fact that the required statistical power was not achieved (i.e., fewer participants than indicated by the sample size calculation), the authors should avoid null hypothesis significance testing (NHST). Instead, the analysis should focus on estimation methods—reporting 95% confidence intervals and unbiased effect sizes (e.g., Cohen's d, also known as Hedges' g)—as well as robust statistics, such as trimmed means and Winsorized standard deviations. Please re-run the analysis accordingly and confirm whether the findings and conclusions remain consistent. Thanks to your open data sharing, I have attached the raw data Excel file along with an example of the results output from Jamovi, which you may use for organizing and presenting your findings.

Cf,

- Estimation statistics: https://pubmed.ncbi.nlm.nih.gov/24220629/

"ESCI" module in Jamovi

- Robust statistics: https://pubmed.ncbi.nlm.nih.gov/31152384/

"Walrus" module in Jamovi

In addition, please generate and replace current figures with estimation plots to display the repeated measures data across time points (at baseline and after creatine supplementation). Cumming or Gardner-Altman estimation plots are recommended (see examples of these figures in the attached Excel file).

Please add the CV% of the instruments/technician to know the variability of all measures.

Answer 10: Thank you for your suggestions on this point, as they prompted us to delve into a subject we were previously unfamiliar with, robust statistics.

We agree that null hypothesis significance testing is a suboptimal approach for our case. Accordingly, we have removed all p-values. Due to the small sample size, trimming-based tests such as Yuen’s paired t-test resulted in a substantial reduction of effective data, leaving us with only 12 pairs of data to analyze.

Instead, given our small sample size, we are proposing to analyze our data by using another robust statistics approach, which uses bootstrap resampling (10 000 replications) to estimate pre-post differences. We chose this approach because it allows us to preserve all observations and avoid strong distributional assumptions. This is complemented by effect

---

## [Editor Report · Decision Letter 1]

Oral creatine in hemodialysis patients increases physical functional capacity and muscle mass, an open label study

PONE-D-25-15109R1

Dear Dr. Basualto-Alarcón,

We’re pleased to inform you that your manuscript has been judged scientifically suitable for publication and will be formally accepted for publication once it meets all outstanding technical requirements.

Kind regards,

**Prof. Diego A. Bonilla**

Academic Editor

PLOS ONE

Additional Editor Comments (optional):

Dear Authors,

Thank you for addressing the revisions to enhance the manuscript's scientific robustness and transparency.

To further strengthen the paper, please consider these final suggestions:

- In "Data Analysis Procedures", add a brief statement clarifying how the robust analytical approach mitigated the limitations of not reaching the a priori sample size target.

- Revise some references in the discussion:

RE, "Kim HJ, et al. Amino Acids. 2011;40:1409–18." with "Kreider RB, et al. J Int Soc Sports Nutr. 2025;22(sup1):2488937. doi:10.1080/15502783.2025.2488937, PMID 40198156"

In line 338: Highlight creatine’s role as a conditionally essential nutrient across the lifespan, citing: Kreider RB, et al. Front Nutr. 2025;12:1578564. doi:10.3389/fnut.2025.1578564, PMID 40331098 as well as the already cited paper by Post A, Tsikas D, Bakker SJL (2019).
---

## [Editor Report · Acceptance letter]

PONE-D-25-15109R1

PLOS ONE

Dear Dr. Basualto-Alarcón,

I'm pleased to inform you that your manuscript has been deemed suitable for publication in PLOS ONE. Congratulations! Your manuscript is now being handed over to our production team.

Kind regards,

on behalf of

Prof. Diego A. Bonilla

Academic Editor

PLOS ONE